# Anti-inflammatory activity of lefamulin versus azithromycin and dexamethasone *in vivo* and *in vitro* in a lipopolysaccharide-induced lung neutrophilia mouse model

Michael Hafner[1], Susanne Paukner[1], Wolfgang W. Wicha[1], Boška Hrvačić[2], Matea Cedilak[2], Ivan Faraho[2], Steven P. Gelone[3]*

1 Nabriva Therapeutics GmbH, Vienna, Austria, 2 Fidelta Ltd, Zagreb, Croatia, 3 Nabriva Therapeutics US, Inc., Fort Washington, Pennsylvania, United States of America

* Steve.Gelone@nabriva.com

## Abstract

Several antibiotics demonstrate both antibacterial and anti-inflammatory/immunomodulatory activities and are used to treat inflammatory pulmonary disorders. Lefamulin is a pleuromutilin antibiotic approved to treat community-acquired bacterial pneumonia (CABP). This study evaluated lefamulin anti-inflammatory effects *in vivo* and *in vitro* in a lipopolysaccharide-induced lung neutrophilia model in which mouse airways were challenged with intranasal lipopolysaccharide. Lefamulin and comparators azithromycin and dexamethasone were administered 30min before lipopolysaccharide challenge; neutrophil infiltration into BALF and inflammatory mediator induction in lung homogenates were measured 4h postchallenge. Single subcutaneous lefamulin doses (10–140mg/kg) resulted in dose-dependent reductions of BALF neutrophil cell counts, comparable to or more potent than subcutaneous azithromycin (10–100mg/kg) and oral/intraperitoneal dexamethasone (0.5/1mg/kg). Lipopolysaccharide-induced pro-inflammatory cytokine (TNF-α, IL-6, IL-1β, and GM-CSF), chemokine (CXCL-1, CXCL-2, and CCL-2), and MMP-9 levels were significantly and dose-dependently reduced in mouse lung tissue with lefamulin; effects were comparable to or more potent than with dexamethasone or azithromycin. Pharmacokinetic analyses confirmed exposure-equivalence of 30mg/kg subcutaneous lefamulin in mice to a single clinical lefamulin dose to treat CABP in humans (150mg intravenous/600mg oral). *In vitro*, neither lefamulin nor azithromycin had any relevant influence on lipopolysaccharide-induced cytokine/chemokine levels in J774.2 mouse macrophage or human peripheral blood mononuclear cell supernatants, nor were any effects observed on IL-8–induced human neutrophil chemotaxis. These *in vitro* results suggest that impediment of neutrophil infiltration by lefamulin *in vivo* may not occur through direct interaction with macrophages or neutrophilic chemotaxis. This is the first study to demonstrate inhibition of neutrophilic lung infiltration and reduction of pro-inflammatory cytokine/chemokine concentrations by clinically relevant lefamulin doses. This anti-inflammatory activity may be beneficial in patients with acute respiratory distress syndrome, cystic fibrosis, or severe inflammation-mediated lung injury, similar to glucocorticoid (eg, dexamethasone) activity. Future lefamulin anti-inflammatory/

**Data Availability Statement:** All relevant data are within the paper and its Supporting Information files.

**Funding:** This research and manuscript development were funded by Nabriva Therapeutics (https://www.nabriva.com), Fort Washington, Pennsylvania, USA, and Vienna, Austria. The funder provided support in the form of salaries for MH, SP, WWW, and SPG, as well as contract funding to Fidelta (Zagreb, Croatia), from which BH, MC, and IF receive salaries. In their respective roles as employees of Nabriva Therapeutics and Fidelta, all authors were involved in aspects of the study design, implementation, decision to publish, and manuscript preparation. The specific roles of these authors are articulated in the "Author Contributions" section. Editorial and medical writing support for manuscript development was provided by Lauriaselle Afanador, PhD, Michael S. McNamara, MS, and Morgan C. Hill, PhD, employees of ICON plc (North Wales, PA, USA), and funded by Nabriva Therapeutics.

**Competing interests:** I have read the journal's policy, and the authors of this manuscript have the following competing interests: MH, SP, WWW, and SPG are employees of/stockholders in Nabriva Therapeutics plc (Dublin, Ireland). BH, MC, and IF are employees of Fidelta (Zagreb, Croatia), which was contracted by Nabriva to conduct the study described in this report. This does not alter our adherence to PLOS ONE policies on sharing data and materials.

immunomodulatory activity studies are warranted to further elucidate mechanism of action and evaluate clinical implications.

## Introduction

Anti-inflammatory and immunomodulatory activities have been observed with various antibiotics, including macrolides (eg, azithromycin), tetracyclines (eg, minocycline), and sulfonamides (eg, trimethoprim-sulfamethoxazole). This has resulted in their use to treat a variety of disorders, including chronic inflammatory conditions (eg, pulmonary and skin disorders), gastrointestinal dysmotility, cancer, and rheumatoid arthritis [1]. In the context of chronic inflammatory pulmonary disorders, substantial evidence supports the use of macrolides to reduce the number of exacerbations and risk of mortality [1]. In cystic fibrosis with *Pseudomonas aeruginosa* infection, long-term azithromycin therapy reduces exacerbations by inhibiting virulence factors in *P. aeruginosa* [2–5]. Macrolides have also been used in non–cystic fibrosis bronchiectasis, diffuse panbronchiolitis, chronic obstructive pulmonary disease, chronic rhinosinusitis, and asthma [1, 2, 6]. In these contexts, the anti-inflammatory effects of macrolides are attributed to reductions in levels of interleukin (IL)-8, neutrophils, neutrophil elastase, and complement 5a and in reduced lymphocyte proliferation.

Inflammation is also observed in acute lung injury and its most severe form, acute respiratory distress syndrome (ARDS), which cause inflammatory damage to the alveolar capillary membrane and excessive uncontrolled pulmonary inflammation [7]. Acute lung injury and ARDS complicate pneumonia and contribute substantially to morbidity and mortality in these patients [7–9]. Systemic inflammation associated with ARDS is commonly treated for up to 3 weeks with corticosteroids such as methylprednisolone or dexamethasone [10]. Macrolide treatment has also been shown to improve clinical outcomes in patients with ARDS, accompanied by evidence that these effects resulted from the immunomodulatory, not the antimicrobial, effects of the macrolides [5, 11]. Lipopolysaccharide (LPS) administration can induce pathologic and biological changes similar to those seen in ARDS, including neutrophilic infiltration and increased intrapulmonary cytokines, which have been extensively studied in experimental models of acute lung injury [12]. Further, research has suggested that increased neutrophil recruitment to the lungs may contribute to tissue damage, particularly in chronic diseases [13].

Pleuromutilin antibiotics inhibit bacterial protein synthesis by binding to the peptidyl transferase center of the 50S ribosomal subunit [14], and lefamulin is the first pleuromutilin antibiotic approved for intravenous (IV) and oral use in humans [15]. Lefamulin has demonstrated potent *in vitro* activity against the pathogens that most commonly cause community-acquired bacterial pneumonia (CABP) [16–19] and, based on the results of two phase 3 clinical trials [20, 21], is approved in the United States and has received a positive opinion from the Committee for Medicinal Products for Human Use in the European Union for the treatment of adults with CABP [15]. The current investigations evaluated the anti-inflammatory effects of lefamulin in LPS-induced lung neutrophilia using *in vivo* and *in vitro* models. The antibiotic azithromycin and the glucocorticoid dexamethasone were included as comparators because of their known anti-inflammatory properties [1, 10].

## Materials and methods

### Pharmacokinetics of lefamulin and azithromycin treatment

Pharmacokinetic parameters were evaluated as previously described [22]. Briefly, female BALB/c mice (weight ~20 g, n = 3/time point; Charles River Deutschland GmbH, Sulzfeld,

Germany) received a single subcutaneous (SC) injection of 35 mg/kg lefamulin (dissolved in 0.9% saline [Sigma-Aldrich Chemie GmbH]) or 35 mg/kg azithromycin (Pfizer Inc, New York, NY, USA; solubilized in 10% dimethyl sulfoxide [DMSO; Sigma-Aldrich Chemie GmbH]). Plasma and bronchoalveolar lavage fluid (BALF) samples were collected at 0.08, 0.25, 0.5, 0.75, 1.5, 3, 6, and 24 hours after lefamulin or azithromycin administration. Plasma samples were analyzed by liquid chromatography with triple quadrupole mass spectrometry (Q Exactive Plus; Thermo Fisher Scientific). Lefamulin and azithromycin concentrations in the epithelial lining fluid (ELF) were calculated from the BALF-to-plasma urea concentration ratio for samples collected at the same time point (BioAssay QuantiChrom™; Thermo Fisher Scientific). These experiments were conducted in Vienna, Austria, according to European Union directive 2010/63/EU and national legislation (GZ:461104/2018/13) regulating the use of laboratory animals in scientific research.

## Measurement of LPS-induced neutrophils and cytokine/chemokine levels in murine lungs

**Animals.** These experiments were conducted in Zagreb, Croatia, according to European Union directive 2010/63/EU and national legislation (Official Gazette 55/13) regulating use of laboratory animals in scientific research, with oversight from an Institutional Committee on Animal Research Ethics (CARE-Zg), and all efforts were made to minimize suffering. Six-week-old male BALB/c mice (Charles River, Calco, Italy) were singly housed in a temperature-controlled (22°C±2°C) environment with a 12:12-hour light:dark cycle and free access to food and water. Mice were given ≥7 days for acclimation before all procedures. The study did not control for additional potential confounders. One day before the start of experimental procedures, all animals were randomized into 6 groups ($n$ = 8/group). Animal group assignment was not blinded.

**Reagents.** LPS lyophilized powder (2 mg; Sigma-Aldrich Chemie GmbH, Munich, Germany) from *Escherichia coli* (O111:B4) was dissolved in 10 mL cold saline (0.9%; Pliva, Zagreb, Croatia), vortexed, and further diluted by mixing 2 mL of this solution with 2 mL saline to reach a final concentration of 5 μg LPS/50 μL saline. Dexamethasone (Sigma-Aldrich Chemie GmbH) was dissolved in carboxymethylcellulose (0.5% in water; Sigma-Aldrich Chemie GmbH) and dosed in a volume of 10 mL/kg per mouse. Lefamulin (BC-3781.Ac; Nabriva Therapeutics, Vienna, Austria) was weighed considering the "as is" purity of 89.3% and dissolved in 0.9% saline. For the lefamulin dose groups (free base: 10, 30, 35, 70, 100, and 140 mg/kg), corresponding lefamulin concentrations were 1.1, 3.4, 3.9, 7.8, 11.2, and 15.7 mg/mL (free base: 1.0, 3.0, 3.5, 7.0, 10.0, and 14.0 mg/mL), respectively. Azithromycin (BC-1024; Nabriva Therapeutics) was weighed using a correction factor of 1.08 and dissolved in 0.5% methylcellulose with 1.25 μL of 1 M citric acid (Alfa Aesar, Ward Hill, MA, USA) for each milligram of azithromycin. For the azithromycin dose groups (free base: 10, 30, and 100 mg/kg), corresponding azithromycin concentrations were 1.1, 3.2, and 10.8 mg/mL (free base: 1.0, 3.0, and 10.0 mg/mL). Lefamulin and azithromycin were each dosed in a volume of 10 mL/kg per mouse. Ketamine hydrochloride (Narketan 10) was acquired from Vetoquinol (Bern, Switzerland), and xylazine hydrochloride (Rompun, 2%) was acquired from Bayer (Leverkusen, Germany).

**Induction of lung neutrophilia and treatments.** Before the LPS challenge, dexamethasone (1 mg/kg intraperitoneal [IP] at 30 minutes before challenge or 0.5 mg/kg oral at 60 minutes before challenge) was administered; vehicle (0.9% saline), lefamulin, and azithromycin were administered SC. The dexamethasone doses and administration routes used (0.5 mg/kg oral or 1 mg/kg IP) produce reproducible effects in the LPS-induced neutrophilia model [23,

24]. Immediately before the LPS challenge, mice were anesthetized via IP injection of ketamine (2 mg/mouse) and xylazine (0.08 mg/mouse). To induce pulmonary neutrophilia, mice in the control group received intranasal (IN) 50 μL saline, and all other animals received 5 μg LPS/ 50 μL saline IN per mouse.

**Bronchoalveolar lavage fluid collection and neutrophil analysis.** Approximately 4 hours after LPS administration, mice were euthanized by an overdose of IP ketamine (200 mg/ kg) and xylazine (16 mg/kg). Tracheostomy was performed, and a Buster cat catheter (1.0 × 130 mm; Kruuse, Langeskov, Denmark), shortened to 3 cm, was clamped into the trachea. The lungs were washed 3 times with cold phosphate-buffered saline (PBS; Sigma-Aldrich Chemie GmbH) in a total volume of 1 mL (0.4, 0.3, and 0.3 mL). The collected BALF samples were centrifuged ($1303 \times g$, 5 min, 4˚C), and resulting cell pellets were each resuspended in 600 μL PBS. Samples were immediately analyzed for total and differential neutrophil cell counts via automated hematology analyzer (XT-2000iV; Sysmex, Kobe, Japan).

**Lung tissue sampling.** After bronchoalveolar lavage, lungs were removed, weighed, snap frozen, and stored at −80˚C until analysis. To homogenize the tissue, the frozen lungs were thawed and placed in Precellys CK28 Hard Tissue tubes (BERTIN Instruments, Montigny-le-Bretonneux, France) in 1 mL PBS (Gibco, Life Technologies, Paisley, UK) supplemented with protease inhibitors (Halt™ Protease Inhibitor Cocktail [100×]; Thermo Fisher Scientific, Rockford, IL, USA). A Precellys homogenizer (BERTIN Instruments) was used, shaking at 6800 rpm for three 30-second pulses separated by 15-second pauses. After homogenization, samples were centrifuged at $18,000 \times g$ for 10 minutes at 4˚C, and the supernatants were collected for analysis.

**Measurement of TNF-α, IL-6, GM-CSF, CXCL-1, CXCL-2, and CCL-2 in lung tissue.** Concentrations of tumor necrosis factor (TNF)-α, IL-6, granulocyte-macrophage colony-stimulating factor (GM-CSF), chemokine (C-X-C motif) ligand (CXCL)-1, CXCL-2, and chemokine (C-C motif) ligand (CCL)-2 in mouse lung homogenates were analyzed with the Mouse Premixed Multi-Analyte Kit (R&D Systems, Minneapolis, MN, USA) per the manufacturer's protocol and using a Luminex 200 System (Luminex Corporation, Austin, TX, USA). Cytokine and chemokine concentrations were determined from blank-corrected median fluorescence intensity of each sample via xPONENT® software (Luminex Corporation), interpolating from standard curves generated with a 5-parameter logistic curve-fit.

**Measurement of MMP-9 and IL-1β in lung tissue.** Mouse Total MMP (matrix metalloprotease)-9 and IL-1β/IL-1F2 DuoSet ELISA (enzyme-linked immunosorbent assay) kits (R&D Systems) were used to measure concentrations of MMP-9 and IL-1β, respectively, in lung homogenates per the manufacturer's protocols. Absorbance at 450 nm was measured using the SpectraMax i3 (Molecular Devices, San Jose, CA, USA). Concentrations of MMP-9 and IL-1β in samples were determined by interpolation from standard curves.

## Chemotaxis of IL-8–activated human neutrophils

**Neutrophil isolation from human blood.** Buffy coat (blood sample fraction comprising white blood cells and platelets) from whole blood was obtained from a healthy adult volunteer at the Croatian Institute of Transfusion Medicine (CITM; Zagreb, Croatia). The CITM ethics committee approved the blood collection process, and the volunteer provided written informed consent prior to blood collection. Neutrophils were isolated from the buffy coat, an aliquot of which was used for cell counting via hematologic analyzer (X500i; Sysmex). The remainder of the buffy coat (7 mL) was diluted with 5 mL of 3% dextran (GE Healthcare, Chicago, IL, USA) and 1.5 mL of 0.18% glucose in PBS (Sigma-Aldrich Chemie GmbH). The diluted buffy coat was drawn into sterile 50-mL syringes and incubated for 30 minutes at RT.

Following incubation, the upper layer of leukocyte-rich plasma (35 mL) was decanted into a new sterile 50-mL conical centrifuge tube, carefully layered onto 15 mL of Lymphoprep (Axis-Shield Diagnostics, Ltd., Dundee, UK) and centrifuged at 400×g (brake turned off) for 35 minutes at RT. The supernatant and mononuclear ring were discarded, and the pellet was resuspended in 10 mL of cold sterile MilliQ water. Erythrocyte lysis was stopped by adding 10 mL of 1.8% saline. Following lysis, neutrophils were centrifuged at 300×g for 5 minutes at RT, supernatants were removed, and neutrophils were resuspended in freshly prepared migration medium (Roswell Park Memorial Institute [RPMI] 1640 medium [Thermo Fisher Scientific] supplemented with 0.05% BSA-FAF [bovine serum albumin-fatty acid free; Sigma-Aldrich Chemie GmbH] and 0.2 μm filtered). Resuspended neutrophils were counted via Sysmex X500i hematologic analyzer. Cell concentrations were adjusted to $5.55 \times 10^6$ neutrophils/mL.

**Preparation of compounds and neutrophil treatment.** All test compounds were reconstituted in DMSO (Sigma-Aldrich Chemie GmbH) to 100-mM stock concentrations, taking into consideration and correcting for salt and purity. A 30-mM stock solution of reference compound Sch527123 (MedChemExpress, Monmouth Junction, NJ, USA) was diluted to 10 mM with DMSO.

Using DMSO as the diluent, 7 consecutive 3.16-fold dilutions were prepared from stock solutions for all compounds. Final working dilutions for all compounds were then prepared by further diluting the compounds 100-fold in migration medium (2 μL of compound was transferred to 198 μL of medium). DMSO was used as vehicle. Lefamulin and azithromycin were tested at concentrations of 0.03, 0.1, 0.3, 1, 3, 10, 30, and 100 μM. The reference compound was tested at concentrations of 0.003, 0.01, 0.03, 0.1, 0.3, 1, 3, and 10 μM. All compounds were tested in triplicate, and the final DMSO concentration was 0.1% per well.

To each well of a 96-well U-bottom plate, 180 μL of resuspended neutrophils ($5.55 \times 10^6$ neutrophils/mL) was added. Test and reference compound working solutions were then added (20 μL/well) for a 10-fold dilution to final testing concentrations. Compounds and cells were mixed by gentle pipetting, and the plate was pre-incubated for 30 minutes at RT with gentle mixing by pipetting.

**Chemotaxis assay and cytotoxicity evaluation.** A 96-well transwell plate insert (Corning Life Sciences, Tewksbury, MA, USA) with donor (upper) wells was removed, and the receiver (lower) wells were filled with 180 μL of 22.22 ng/mL rhIL-8 solution in migration medium. Negative control wells received 180 μL of migration medium alone. To all wells, 20 μL of compound working solutions or DMSO vehicle was added. Once pre-incubation of cells with compounds was complete, the transwell inserts were carefully placed on the receiver plate, with caution to avoid air bubble formation, which could prevent cell migration. To the upper wells, 75 μL of cell suspension (375,000 cells/well) was added, taking care not to tear the membrane with pipette tips.

The plate was incubated in a $CO_2$ incubator (37˚C, 5% $CO_2$, 95% humidity) for 1 hour. After incubation, the number of cells that migrated to the lower wells was quantified by removing the Transwell insert and adding 200 μL of CellTiter Glo reagent (CellTiter-Glo Luminescent Cell Viability Assay, Promega, Madison, WI, USA) to each lower well to quantify the amount of adenosine triphosphate (ATP) present, signaling the presence of metabolically active cells. The plate was incubated for 10 minutes in the dark at RT, followed by transferring 150 μL of solution into wells of a white 96-well plate (Lumitrac 200; Greiner Bio-One International GmbH, Kremsmunster, Austria). Luminescence was measured by use of EnVision 2104 Multilabel Plate Reader (PerkinElmer, Waltham, MA, USA), with exposition time of 0.1 seconds.

To evaluate cytotoxicity of human neutrophils, cells that were not used for the chemotaxis assay were incubated with compounds for an additional hour in the $CO_2$ incubator (37˚C, 5%

$CO_2$, 95% humidity), after which 125 μL of CellTiter Glo reagent was added to each well, and the plate was incubated for 10 minutes in the dark at RT. The solution (150 μL) was transferred into wells of a white 96-well plate (Lumitrac 200), and luminescence was measured by use of EnVision 2104 Multilabel Plate Reader, with an exposition time of 0.1 seconds.

### Statistical analyses

The pharmacokinetic profiles of lefamulin and azithromycin were analyzed by the sparse sampling noncompartmental method (Phoenix WinNonlin 6, Certera, Princeton, NJ, USA) based on the nominal time points. Area under the curve (AUC) values were determined using the linear trapezoidal method.

For the lung neutrophilia analyses, statistical analyses were performed using GraphPad Prism (versions 5.04 and 8.1.1; GraphPad Software, Inc., La Jolla, CA, USA). Differences between treated versus vehicle groups were determined using the Mann-Whitney test and were considered statistically significant when $P<0.05$. Outliers in the analysis of cytokine and chemokine concentrations were identified using the Grubbs test. No criteria were set for inclusion/exclusion of animals during the experiment or data points during the analysis.

For the chemotaxis and cytotoxicity evaluations, average relative light unit values were calculated from all untreated vehicle samples and, for each sample, percentage of vehicle value was calculated. The tested compound was considered cytotoxic if reduction from untreated vehicle was $\geq$20%.

Additional Methods can be found in **S1 Appendix**.

## Results

### Pharmacokinetic analysis of lefamulin in mice

The pharmacokinetics of a single SC injection of 35 mg/kg lefamulin or azithromycin was assessed in mice. Plasma AUC from time 0 to 24 hours ($AUC_{0-24h}$) values after 35 mg/kg SC lefamulin or azithromycin were 6.25±0.93 or 11.6±1.37 μg·h/mL, respectively (**Table 1**). In terms of distribution from plasma to ELF in mice, both lefamulin and azithromycin showed rapid penetration into the lung compartment following a single SC dose, with comparable times to maximum concentration for both matrices. For each drug, the AUC ratio for ELF to plasma was approximately 2-fold.

### Effects of lefamulin on LPS-induced lung neutrophilia in mice

In the *in vivo* mouse lung neutrophilia model, lefamulin treatment at doses of 10, 30, and 100 mg/kg SC at 30 minutes before intranasal LPS challenge (30 minutes pretreatment) was associated with a dose-dependent reduction in total cell and neutrophil recruitment to the lungs

**Table 1. Pharmacokinetic profile in plasma and epithelial lining fluid of mice following a single subcutaneous dose of lefamulin or azithromycin 35 mg/kg.**

| | Matrix | $t_{max}$ (h), | $C_{max}$ (μg/mL), | $AUC_{0-24h}$ (μg·h/mL), | $AUC_{0-24h}$/Dose (μg·h/mL)/(mg/kg), |
|---|---|---|---|---|---|
| | | mean | mean±SEM | mean±SEM | mean |
| **Lefamulin** | Plasma | 0.50 | 1.33±0.18 | 6.25±0.93 | 0.18 |
| | ELF | 0.50 | 2.16±0.50 | 12.6±1.17 | 0.36 |
| **Azithromycin** | Plasma | 0.08 | 2.33±0.83 | 11.6±1.37 | 0.33 |
| | ELF | 0.25 | 2.35±1.45 | 26.7±5.78 | 0.76 |

$AUC_{0-24h}$ = area under the curve from time 0 to 24 hours; $C_{max}$ = maximum observed plasma concentration; ELF = epithelial lining fluid; $t_{max}$ = time of maximum observed concentration.

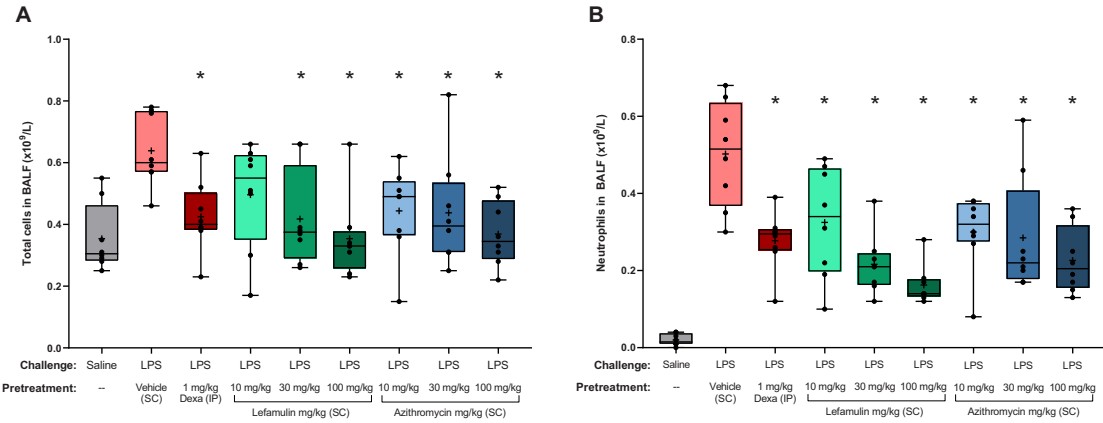

**Fig 1.** (A) Total and (B) neutrophil cell counts in BALF after LPS induction. BALF, bronchoalveolar lavage fluid; Dexa, dexamethasone; IP, intraperitoneal; LPS, lipopolysaccharide; SC, subcutaneous. Box and whisker plots show 25% percentile, median, and 75% percentile in box, with minimum and maximum values shown with whiskers. Means are shown with "+" and raw data points with black circles. *$P<0.05$ vs LPS/vehicle via Mann-Whitney test.

(measured in BALF) at 4 hours postchallenge compared with the vehicle control group (no treatment; **Fig 1**). These reductions were comparable to those observed following treatment with 1 mg/kg IP dexamethasone, with a tendency toward more potent inhibition of total cell counts and neutrophil cell counts at the highest lefamulin dose of 100 mg/kg. Pretreatment with azithromycin at doses of 10, 30, and 100 mg/kg SC demonstrated significant dose-dependent reductions in total cell and neutrophil counts as well, although the effects on neutrophil counts in BALF were more pronounced with the 30- and 100-mg/kg lefamulin doses. Similar results were observed in an independent experiment using higher doses of lefamulin and azithromycin (35, 70, and 140 mg/kg; **S1 Fig in S1 Appendix**).

### Effects of lefamulin on cytokine production *in vivo* and *in vitro*

To assess the effects of lefamulin, azithromycin, and dexamethasone on LPS-induced pro-inflammatory cytokines, chemokines, and MMP-9, lungs from the mouse model of neutrophilia were homogenized and evaluated via Luminex immunoassay and ELISA. As observed with dexamethasone (1 mg/kg IP), lefamulin (10, 30, and 100 mg/kg SC) was generally associated with significantly reduced levels of all cytokines and chemokines assessed, as well as of MMP-9 (**Fig 2**). TNF-α and IL-6 concentrations were significantly reduced at all lefamulin doses tested compared with vehicle control. The reductions observed with lefamulin were similar to those observed with 1 mg/kg IP dexamethasone. In contrast, azithromycin was associated with significant reductions in TNF-α concentrations at 10 and 30 mg/kg, with no significant effect observed with 100 mg/kg, and IL-6 levels were reduced to a lesser extent with azithromycin than with lefamulin or dexamethasone.

A dose-dependent effect on IL-1β concentrations was also observed with lefamulin; however, significant inhibition of IL-1β was observed only with the highest lefamulin dose (100 mg/kg), similar to that observed with dexamethasone 1 mg/kg. In contrast, azithromycin reduced IL-1β concentrations at all doses, although only the lowest dose showed a significant reduction similar to that of 100 mg/kg lefamulin. Significant reductions in MMP-9 levels were observed with 30 and 100 mg/kg lefamulin, with effects similar to those seen with dexamethasone, whereas reductions in MMP-9 levels with azithromycin (all doses) were not as pronounced as with lefamulin. Significant reductions in chemokines and GM-CSF were also

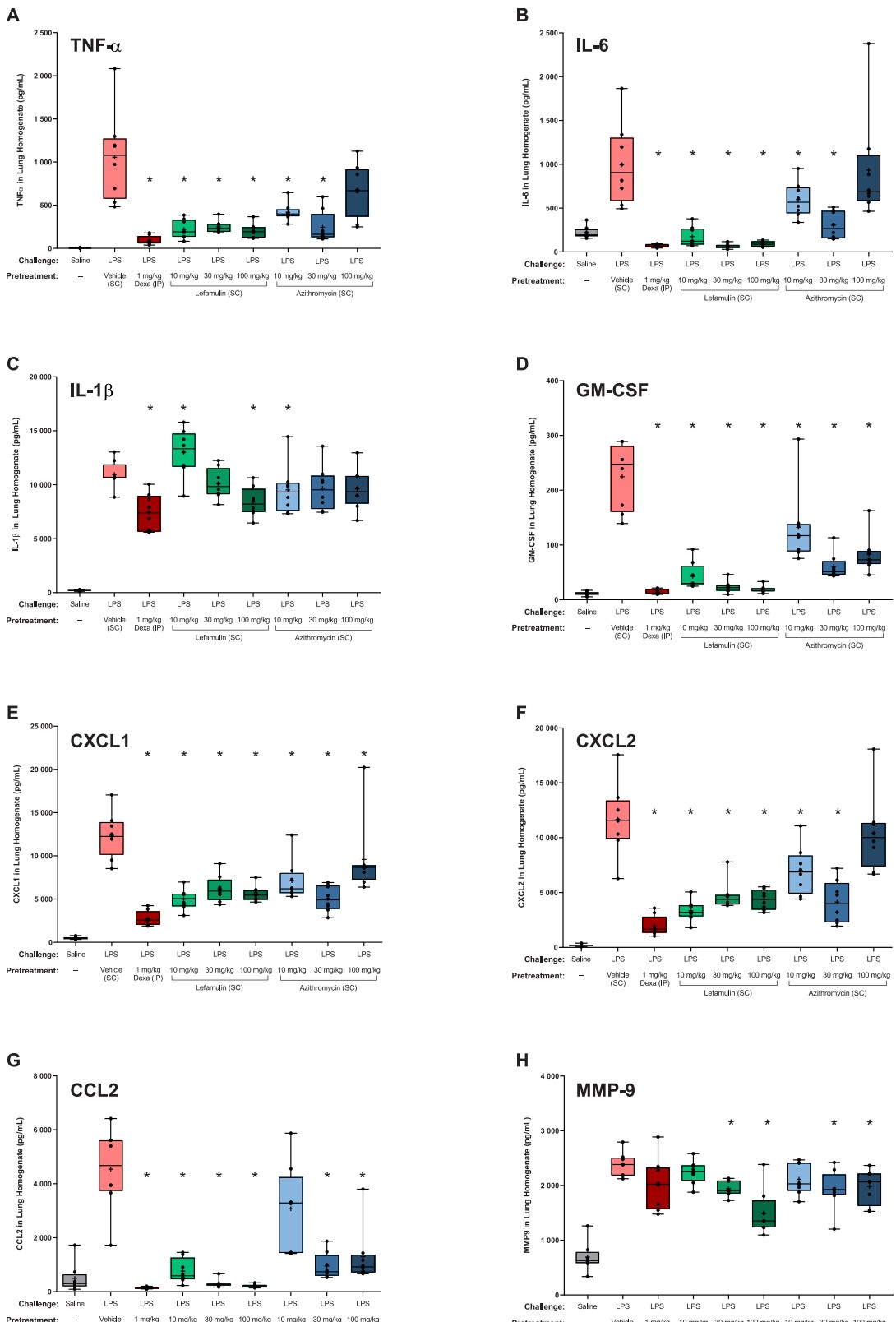

**Fig 2. LPS-induced cytokines and chemokines in mouse lung homogenate.** CCL, chemokine (C-C motif) ligand; CXCL, chemokine (C-X-C motif) ligand; Dexa, dexamethasone; GM-CSF, granulocyte-macrophage colony-stimulating factor; IL,

interleukin; IP, intraperitoneal; LPS, lipopolysaccharide; MMP, matrix metalloprotease; SC, subcutaneous; TNF, tumor necrosis factor. Box and whisker plots show 25% percentile, median, and 75% percentile in box, with minimum and maximum values shown with whiskers. Means are shown with "+" and raw data points with black circles. *$P<0.05$ vs LPS/vehicle via Mann-Whitney test.

observed with all lefamulin doses and dexamethasone, whereas reductions with azithromycin were significant but not as pronounced as those with lefamulin or dexamethasone.

To investigate further the mechanism of inhibition of LPS-induced pulmonary neutrophilia by lefamulin and azithromycin, cell viability and LPS-induced production of cytokines, chemokines, and MMP-9 were assessed *in vitro* in human neutrophils, J774.2 mouse macrophages, and human peripheral blood mononuclear cells (PBMCs). Cells were pretreated with lefamulin and azithromycin similarly as described for the *in vivo* experiment. In J774.2 cells, cell viability was reduced at 30 μM (15.2 μg/mL free base) and 100 μM (50.8 μg/mL free base) lefamulin and at 100 μM (74.9 μg/mL) azithromycin (**S1 Table in S1 Appendix**). In PBMCs, cell viability was reduced at 100 μM lefamulin, and no cytotoxic effects were observed with azithromycin ≤100 μM. Based on these results, data in J774.2 cells are not shown for the 30- and 100-μM doses of lefamulin and azithromycin, and data in PBMCs are not shown for 100 μM lefamulin or azithromycin.

Although treatment with dexamethasone resulted in dose-dependent reductions of TNF-α, IL-6, and IL-1β in J774.2 mouse macrophages (**S2 Fig in S1 Appendix**) and of all measured cytokines and chemokines in human PBMCs (**S3 Fig in S1 Appendix**), little to no reduction in levels of the measured LPS-induced cytokines, chemokines, or MMP-9 was observed in supernatants from either cell type at the concentrations of lefamulin or azithromycin tested. In J774.2 macrophages, however, IL-6 and IL-1β levels showed a trend toward reduction with lefamulin (**S2 Fig in S1 Appendix**).

### Effects of lefamulin and azithromycin on neutrophilic chemotaxis

The effects of lefamulin and azithromycin on IL-8–induced chemotaxis of human neutrophils were also assessed. Treatment with the reference compound Sch527123 (an antagonist of chemokine [C-X-C motif] ligand [CXCL]-1 and CXCL-2) resulted in dose-dependent inhibition neutrophilic chemotaxis (**Fig 3**), whereas treatment with 0.03 to 30 μM lefamulin (0.02–15.2 μg/mL free base) or azithromycin (0.02–22.5 μg/mL) had no effect on IL-8–induced chemotaxis of human neutrophils. In neutrophils, cell viability was reduced at 100 μM (50.8 μg/mL free base) lefamulin to 86% of vehicle control (**S1 Table in S1 Appendix**), whereas no cytotoxic effects were observed at 30 μM lefamulin or azithromycin ≤100 μM. Based on these results, data in neutrophils are not shown for 100 μM lefamulin or azithromycin.

### Discussion

Previous research has demonstrated anti-inflammatory and immunomodulatory effects with antibiotics, including macrolides, tetracyclines, and fluoroquinolones, but for many, the higher dose and/or longer duration of antibiotic treatment that may be required for anti-inflammatory effects (compared with those required for anti-infective effects) must be balanced with both adverse effects and the risk of emerging microbial resistance [1]. In the context of acute infection, the timing of anti-inflammatory treatment may also need to be considered, as a required anti-pathogen immune response can be hindered by an anti-inflammatory treatment effect [25]. Therefore, an antibiotic able to exert anti-inflammatory effects at anti-infective doses would be of interest, particularly in chronic inflammatory pulmonary disorders or ARDS, in which neutrophilic lung infiltration is a key characteristic [7].

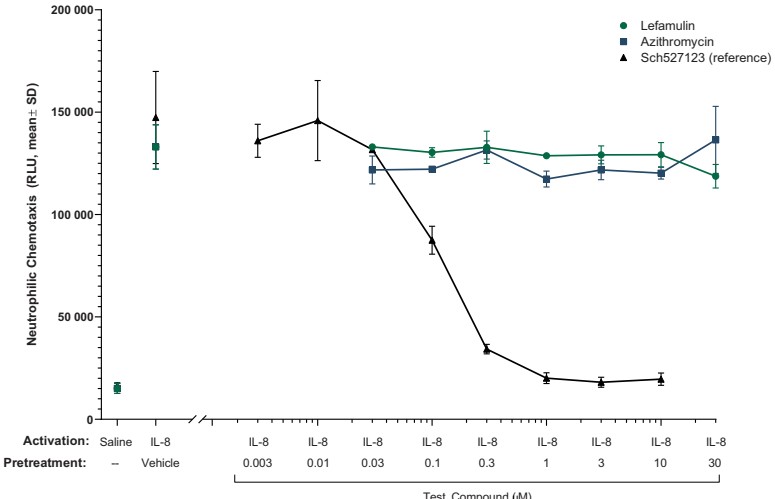

**Fig 3. Chemotaxis of IL-8–activated human neutrophils with lefamulin, azithromycin, and Sch527123 (reference compound).** IL, interleukin; RLU, relative light unit; SD, standard deviation.

To our knowledge, these analyses are the first to investigate the anti-inflammatory activity of the pleuromutilin antibiotic lefamulin. In a mouse model of lung neutrophilia, pretreatment with lefamulin at doses of 10, 30, and 100 mg/kg SC was associated with almost complete reduction in LPS-induced recruitment of total cells and neutrophils to the lungs at 4 hours postchallenge compared with the vehicle control group (no treatment). Similar effects were observed with the same SC doses of the macrolide antibiotic azithromycin (although the exposures associated with these doses of azithromycin exceed those of the corresponding human clinical doses), as well as with 1 mg/kg IP dexamethasone, a known anti-inflammatory glucocorticoid. Levels of cytokines, chemokines, and MMP-9 in lung homogenates were reduced following pretreatment with lefamulin, dexamethasone, or azithromycin, although the reductions associated with 10 to 100 mg/kg lefamulin tended to be comparatively larger in magnitude and more consistent than those associated with 10 to 100 mg/kg azithromycin. The results observed here were generally consistent with previous anti-inflammatory effects seen with azithromycin [26–28], although some inconsistencies exist that may be related to a variety of methodologic differences between the studies, including LPS dose, azithromycin dose, and route of azithromycin administration.

The plasma $AUC_{0-24h}$ observed here for SC lefamulin was comparable to exposures seen in healthy volunteers following a single dose of 150 mg IV lefamulin [29], which is half of the daily dose approved for the treatment of CABP. Our data are also consistent with lefamulin pharmacokinetics previously described in mice administered 35 and 70 mg/kg SC lefamulin [22], which showed dose-AUC linearity; hence, a daily dose of 70 mg/kg in mice provides equivalent exposure to 300 mg IV lefamulin in humans, which is the recommended daily dose to treat CABP [15]. Thus, the *in vivo* data presented here (with lefamulin doses of 10–100 mg/kg) indicate that lefamulin anti-inflammatory activity is seen with a plasma exposure that is lower than that achieved at the corresponding antimicrobial clinical dose (ie, at subtherapeutic concentrations). The plasma exposure observed here following a 35-mg/kg SC azithromycin dose (11.6 μg·h/mL), however, was 2-fold higher than that following a 500-mg IV 3-hour infusion in healthy volunteers (5.0 μg·h/mL) [30]. According to the label, the daily AUC following the first and fifth daily doses of 500 mg IV azithromycin showed only an 8% increase in maximum observed plasma concentration but a 61% increase in AUC [30]. These data suggest that

azithromycin anti-inflammatory activity is seen with a plasma exposure that is achieved at doses higher than the approved daily clinical dose of 500 mg IV.

The robust anti-inflammatory effects observed within a range of clinically exposure-equivalent lefamulin doses (eg, 10–100 mg/kg) in this mouse model of neutrophilic inflammation suggest that lefamulin inhibits either LPS-induced pro-inflammatory signaling, resulting in reduced neutrophil accumulation, or alternatively, directly inhibits neutrophil infiltration into the lung, resulting in reduced levels of neutrophil-contributed cytokines and chemokines. Both potential mechanisms are consistent with the dose-dependent anti-inflammatory effects observed *in vivo* following lefamulin pretreatment.

LPS activates macrophages and monocytes to release high amounts of pro-inflammatory cytokines, chemokines, and MMPs via toll-like receptor (TLR)-2 and TLR-4 signaling [31, 32]. To investigate whether lefamulin targets macrophages and monocytes, *in vitro* experiments were performed using LPS-activated J774.2 mouse macrophages. To investigate whether any effect was species specific, human PBMCs were also tested.

In contrast to the *in vivo* results, pretreatment of mouse macrophages and human PBMCs with lefamulin at doses of 0.03 to 10 μM had little to no effect on the levels of LPS-induced cytokines, chemokines, or MMP-9 in cell supernatants. At the azithromycin doses tested, these results were consistent with previous studies [27, 28]. We therefore concluded that macrophages/monocytes are not the target cells of lefamulin, or alternatively, the *in vitro* experiments do not appropriately mimic the *in vivo* situation.

As macrophages seemed to be unaffected by lefamulin, neutrophils were also evaluated as potential target cells. To examine possible effects of lefamulin on neutrophil recruitment to the lungs, lefamulin and azithromycin were tested in a model of IL-8–induced neutrophil chemotaxis. However, doses of 0.03 to 30 μM lefamulin or azithromycin did not affect IL-8–induced chemotaxis of human neutrophils. These findings therefore suggest that, although lefamulin may impede neutrophil chemotaxis based on the *in vivo* results, it does not do so by direct interaction with macrophages or neutrophils.

Other potential target cells of lefamulin are endothelial cells and type 2 pneumocytes. When activated by pro-inflammatory mediators such as TNF-α and IL-1, the endothelium becomes a major participant in the generation of the inflammatory response, increasing adhesion molecules and enabling leukocyte extravasation [33]. Type 2 pneumocytes also have TLRs and secrete cytokines and chemokines following activation by bacterial components like LPS [34]. However, the role of these cells in the lefamulin-induced anti-inflammatory effect is not yet known.

Notably, compared with the other cytokines and chemokines tested, significant reductions in levels of IL-1β, a key pro-inflammatory cytokine [35], were observed only at higher lefamulin concentrations. This may suggest that the anti-inflammatory effects observed with lefamulin proceed via a different mechanism than that of azithromycin and other macrolides, which inhibit the production of IL-1β by alveolar macrophages [27, 28]; the possibility that lefamulin acts via a different anti-inflammatory mechanism from azithromycin is further supported by the inability of lefamulin to reduce LPS-induced cytokines, chemokines, or MMP-9 *in vitro* in J774.2 mouse macrophages or human PBMCs.

In earlier investigations, the veterinary pleuromutilin antibiotic valnemulin showed *in vitro* and *in vivo* anti-inflammatory effects [36, 37]. LPS-induced pulmonary edema, accumulation of inflammatory cells in BALF (eg, neutrophils and macrophages), and increased inflammatory cytokines (eg, TNF-α and IL-6) were significantly attenuated in mice pretreated with valnemulin or dexamethasone compared with no-treatment control group, with histologic analysis suggesting a protective effect from valnemulin on LPS-induced acute lung injury [36]. Valnemulin treatment in murine RAW 264.7 macrophages also significantly inhibited LPS-

induced production of inflammatory mediators, including nitric oxide, prostaglandin $E_2$, TNF-$\alpha$, and IL-6 [37]. Likewise, significantly reduced TNF-$\alpha$, IL-6, and CCL-2 serum levels were observed in a methicillin-resistant *Staphylococcus aureus* wound infection mouse model following treatment with amphenmulin, a pleuromutilin derivative currently in development for veterinary use [38]; due to the model used, however, further studies are needed to determine if these effects were the direct result of anti-inflammatory activity, an indirect consequence of antimicrobial activity, or both.

Although further research into the anti-inflammatory mechanism of lefamulin is needed, these findings have clinical implications. Glucocorticoids such as dexamethasone have demonstrated benefits in reducing mortality in patients with severe inflammation-mediated lung injury [25]. Likewise, the immunomodulatory treatment tocilizumab has been shown to reduce mortality in similar patient populations [39, 40]. Inhibition of neutrophilic lung infiltration with lefamulin may be beneficial, for example, during the early phase of ARDS, which is characterized by disruption of alveolar epithelial and endothelial barriers as well as widespread neutrophilic alveolitis, leading to formation of protein-rich edema in interstitium and alveolar spaces [7, 41]. However, these data have some limitations. First, although endotoxin (eg, LPS) models are suitable for assessing acute inflammation and early immune response [42], they do not reproduce exactly the complex pathophysiology of human sepsis or ARDS [43]. Therefore, the findings presented here provide an incomplete picture of the immunomodulatory effects of lefamulin and further research is needed. Second, only a single time point, 4 hours following LPS challenge, was examined in the *in vivo* models; future evaluation of additional time points in this inflammatory response may provide valuable insight. Third, methodologic differences between the analyses presented here and previous studies of azithromycin make comparisons across studies difficult.

In conclusion, these results suggest that lefamulin has anti-inflammatory properties similar to, or more potent than, those of macrolide antibiotics, which are currently used as anti-inflammatory therapy for pulmonary disorders [44]. Like macrolides, lefamulin inhibits bacterial protein synthesis and demonstrates excellent tissue penetration, accumulation in macrophages, and immunomodulatory effects (eg, neutrophilic inflammation inhibition) [15, 22, 44, 45]. Moreover, azithromycin or clarithromycin pretreatment in LPS-induced acute lung injury models similarly resulted in significantly reduced neutrophil recruitment [27]. Further research on the anti-inflammatory and immunomodulatory properties of lefamulin and its potential as a treatment for inflammatory lung diseases is warranted.

## Supporting information

**S1 Appendix. Supplementary methods, S1-S3 Figs, S1 Table.**
(PDF)

**S2 Appendix. Full dataset.**
(XLSX)

## Acknowledgments

Editorial and medical writing support for manuscript development was provided by Lauriaselle Afanador, PhD, Michael S. McNamara, MS, and Morgan C. Hill, PhD, employees of ICON (North Wales, PA, USA).

## Author Contributions

**Conceptualization:** Michael Hafner, Susanne Paukner, Wolfgang W. Wicha, Boška Hrvačić, Steven P. Gelone.

**Data curation:** Boška Hrvačić, Matea Cedilak, Ivan Faraho.

**Formal analysis:** Michael Hafner, Susanne Paukner, Wolfgang W. Wicha, Boška Hrvačić, Matea Cedilak, Ivan Faraho, Steven P. Gelone.

**Investigation:** Boška Hrvačić, Matea Cedilak, Ivan Faraho.

**Methodology:** Michael Hafner, Susanne Paukner, Wolfgang W. Wicha, Boška Hrvačić, Matea Cedilak, Ivan Faraho, Steven P. Gelone.

**Resources:** Wolfgang W. Wicha, Boška Hrvačić, Matea Cedilak, Ivan Faraho.

**Writing – review & editing:** Michael Hafner, Susanne Paukner, Wolfgang W. Wicha, Boška Hrvačić, Matea Cedilak, Ivan Faraho, Steven P. Gelone.

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
