## [Decision Letter · Decision Letter 0]

30 Oct 2020

PONE-D-20-24953

Anti-inflammatory activity of lefamulin in a lipopolysaccharide-induced lung neutrophilia model

PLOS ONE

Dear Dr. Gelone,

Thank you for submitting your manuscript to PLOS ONE. After careful consideration, we feel that it has merit but does not fully meet PLOS ONE’s publication criteria as it currently stands. Therefore, we invite you to submit a revised version of the manuscript that addresses the points raised during the review process.

Your paper has been reviewed by two experts in the field. Both reviewers believe that the paper is not strong enough for acceptance for publication. I would like to provide you with an opportunity to revise the manuscript. See below for comments provided by the reviewers. 

Reviewers' comments

•     The abstract lacks a brief introduction for the subject and a detailed methodology and a conclusion.

•     The authors did not introduce well the problem they will talk about and which findings the previous authors did reach in addition to the gap that this study will solve. This must be taken into consideration in the introduction section. Moreover, the authors talk about the immunomodulatory effects of lefamulin without studying them in their manuscript. Finally, the authors must detail the danger of lipopolysaccharide, especially from Escherichia coli as they dealt with. 

•     The manuscript had little data to can confirm the anti-inflammatory activity of lefamulin. This lacks the representativeness of the data. They should add more work on this point, especially gene expression data.

•     The authors compared the effect of lefamulin with that of dexamethasone. This was not clear. Please, provide results about that in all parts in the manuscript in addition to supplying the statistical documentation for this comparison.

•     Majority of the paragraphs of the materials section lacks any references and that makes the research lacks reality. It must be taken in consideration.

•     The results lack the significance data to be reliable.

•     The authors must interpret their results after its comparison with others in the discussion section.

•     The company names for any reagents must be mentioned in the materials and methods section.

We look forward to receiving your revised manuscript.

Kind regards,

Yu-Wei Lin, PhD

Academic Editor

PLOS ONE

Journal Requirements:

"I have read the journal’s policy and the authors of this manuscript have the following competing interests: MH, SP, WWW, and SPG are employees of/stockholders in Nabriva Therapeutics plc (Dublin, Ireland). BH is an employee of Fidelta (Zagreb, Croatia), which was contracted by Nabriva to conduct the study described in this report."

We note that one or more of the authors are employed by a commercial company: Nabriva Therapeutics plc, Fidelta Ltd.

2.1. Please provide an amended Funding Statement declaring this commercial affiliation, as well as a statement regarding the Role of Funders in your study. If the funding organization did not play a role in the study design, data collection and analysis, decision to publish, or preparation of the manuscript and only provided financial support in the form of authors' salaries and/or research materials, please review your statements relating to the author contributions, and ensure you have specifically and accurately indicated the role(s) that these authors had in your study. You can update author roles in the Author Contributions section of the online submission form.

2.2. Please also provide an updated Competing Interests Statement declaring this commercial affiliation along with any other relevant declarations relating to employment, consultancy, patents, products in development, or marketed products, etc.  

Reviewers' comments:

Reviewer's Responses to Questions

**Comments to the Author**

1. Is the manuscript technically sound, and do the data support the conclusions?

Reviewer #1: Partly

Reviewer #2: No

2. Has the statistical analysis been performed appropriately and rigorously? 

Reviewer #1: No

Reviewer #2: Yes

3. Have the authors made all data underlying the findings in their manuscript fully available?

Reviewer #1: Yes

Reviewer #2: Yes

4. Is the manuscript presented in an intelligible fashion and written in standard English?

Reviewer #1: No

Reviewer #2: No

5. Review Comments to the Author

Reviewer #1: The authors evaluated the anti-inflammatory activity of lefamulin in a murine lipopolysaccharide-induced lung neutrophilia model. In spite of the scientific value of the subject, it lacks some important details.

Therefore, it is greatly suggested that the manuscript is not ready to be accepted now. My decision is accept after major revisions. I have several comments listed below.

Reviewer #2: The manuscript has no significant data to prove their hypothesis. lefamulin is a very good antibiotic and can show anti-inflammatory responses. The authors failed to bring any conclusive decisions in it. The minimum dosage of drug used by them was high. The positive control used by them was 0.5 mg/kg. The compound used by them was 35 mg/kg BW. The author failed to show any cytokine quantification.

---

## [Author Response · Author response to Decision Letter 0]

29 Apr 2021

Detailed Response to Reviewers

Journal Requirements/Editors’ Comments:

*Authors Response: We confirm that the manuscript aligns with the journal’s style requirements, including the file name conventions.

"I have read the journal’s policy and the authors of this manuscript have the following competing interests: MH, SP, WWW, and SPG are employees of/stockholders in Nabriva Therapeutics plc (Dublin, Ireland). BH is an employee of Fidelta (Zagreb, Croatia), which was contracted by Nabriva to conduct the study described in this report."

We note that one or more of the authors are employed by a commercial company: Nabriva Therapeutics plc, Fidelta Ltd.

*Authors’ Response: The Competing Interests Statement has been updated as requested and is included in the cover letter.

*Authors’ Response: The Financial Disclosure Statement has been updated as requested and is included in the cover letter.

4. Please also provide an updated Competing Interests Statement declaring this commercial affiliation along with any other relevant declarations relating to employment, consultancy, patents, products in development, or marketed products, etc.

Within your Competing Interests Statement, please confirm that this commercial affiliation does not alter your adherence to all PLOS ONE policies on sharing data and materials by including the following statement: "This does not alter our adherence to PLOS ONE policies on sharing data and materials.” (as detailed online in our guide for authors https://protect-eu.mimecast.com/s/sFj4C868VC6kOWKYiEsxTV?domain=journals.plos.org). If this adherence statement is not accurate and there are restrictions on sharing of data and/or materials, please state these. Please note that we cannot proceed with consideration of your article until this information has been declared.

Please know it is PLOS ONE policy for corresponding authors to declare, on behalf of all authors, all potential competing interests for the purposes of transparency. PLOS defines a competing interest as anything that interferes with, or could reasonably be perceived as interfering with, the full and objective presentation, peer review, editorial decision-making, or publication of research or non-research articles submitted to one of the journals. Competing interests can be financial or non-financial, professional, or personal. Competing interests can arise in relationship to an organization or another person. Please follow this link to our website for more details on competing interests: https://protect-eu.mimecast.com/s/sFj4C868VC6kOWKYiEsxTV?domain=journals.plos.org.

*Authors’ Response: The Competing Interests Statement has been updated as requested and is included in the cover letter.

*Authors’ Response: Descriptions of ethical approval for the studies described in the manuscript are included only in the Methods section. Statements describing approval of the animal studies are provided on page 6 (lines 104‒107 and 111‒115). Statements describing approval for the collection of human blood appear on page 9 (lines 191‒195) and in S1 Appendix, page 3, paragraph 2.

Reviewers' Comments:

1. The abstract lacks a brief introduction for the subject and a detailed methodology and a conclusion.

*Authors’ Response: Thank you for your review. We have revised the Abstract to provide brief background on the anti-inflammatory activity of some antibiotics, lefamulin as an antibiotic, and the study described in the manuscript. The PLOS ONE Instructions to Authors suggest that methodologic details should not be included in the Abstract. Accordingly, we have revised the Abstract to present a combined Methods & Results section that describes basic methodology but focuses on the study results. The Abstract conclusions have also been expanded as requested. The Abstract now reads as follows (pages 2‒3):

Several antibiotics demonstrate both antibacterial and anti-inflammatory/immunomodulatory activities and are used to treat inflammatory pulmonary disorders. Lefamulin is a pleuromutilin antibiotic approved to treat community-acquired bacterial pneumonia (CABP). This study evaluated lefamulin anti-inflammatory effects in vivo and in vitro in a lipopolysaccharide-induced lung neutrophilia model in which mouse airways were challenged with intranasal lipopolysaccharide.

Lefamulin and comparators azithromycin and dexamethasone were administered 30min before lipopolysaccharide challenge; neutrophil infiltration into BALF and inflammatory mediator induction in lung homogenates were measured 4h postchallenge. Single subcutaneous lefamulin doses (10‒140mg/kg) resulted in dose-dependent reductions of BALF neutrophil cell counts, comparable to or more potent than subcutaneous azithromycin (10‒100mg/kg) and oral/intraperitoneal dexamethasone (0.5/1mg/kg). Lipopolysaccharide-induced pro-inflammatory cytokine (TNF-α, IL-6, IL-1β, and GM-CSF), chemokine (CXCL-1, CXCL-2, and CCL-2), and MMP-9 levels were significantly and dose-dependently reduced in mouse lung tissue with lefamulin; effects were comparable to or more potent than with dexamethasone or azithromycin. Pharmacokinetic analyses confirmed exposure-equivalence of 30mg/kg subcutaneous lefamulin in mice to a single clinical lefamulin dose to treat CABP in humans (150mg intravenous/600mg oral). In vitro, neither lefamulin nor azithromycin had any relevant influence on lipopolysaccharide-induced cytokine/chemokine levels in J774.2 mouse macrophage or human peripheral blood mononuclear cell supernatants, nor were any effects observed on IL-8‒induced human neutrophil chemotaxis. These in vitro results suggest that impediment of neutrophil infiltration by lefamulin in vivo may not occur through direct interaction with macrophages or neutrophilic chemotaxis.

This is the first study to demonstrate inhibition of neutrophilic lung infiltration and reduction of pro-inflammatory cytokine/chemokine concentrations by clinically relevant lefamulin doses. This anti-inflammatory activity may be beneficial in patients with acute respiratory distress syndrome, cystic fibrosis, or severe inflammation-mediated lung injury, similar to glucocorticoid (eg, dexamethasone) activity. Future lefamulin anti-inflammatory/immunomodulatory activity studies are warranted to further elucidate mechanism of action and evaluate clinical implications.

2. The authors did not introduce well the problem they will talk about and which findings the previous authors did reach in addition to the gap that this study will solve. This must be taken into consideration in the introduction section. Moreover, the authors talk about the immunomodulatory effects of lefamulin without studying them in their manuscript. Finally, the authors must detail the danger of lipopolysaccharide, especially from Escherichia coli as they dealt with.

*Authors’ Response: The Introduction section (pages 4‒5) has been revised extensively to better define the experimental questions in the context of published literature. The Results section (pages 12–17) has been expanded to present additional mechanistic studies around the immunomodulatory effects of lefamulin. The Discussion section (pages 17–21) has also been revised to interpret these expanded results in the context of the published literature.

Because lipopolysaccharide was used in our study only as a reagent within the context of a well-defined experimental model, details of dangers associated with lipopolysaccharide are outside the scope of this article and have not been added.

3. The manuscript had little data to can confirm the anti-inflammatory activity of lefamulin. This lacks the representativeness of the data. They should add more work on this point, especially gene expression data.

*Authors’ Response: Although gene expression data were not available, the Results section of our manuscript has been expanded to present additional mechanistic studies around the immunomodulatory effects of lefamulin, including in vivo and in vitro measurement of cytokine and chemokine levels, as well as findings from chemotaxis and cell viability analyses.

4. The authors compared the effect of lefamulin with that of dexamethasone. This was not clear. Please, provide results about that in all parts in the manuscript in addition to supplying the statistical documentation for this comparison.

*Authors’ Response: With the new data that have been added to the manuscript, both azithromycin and dexamethasone are now comparators for lefamulin. These comparisons and the associated statistical methodology are now clearly described in the Results (pages 12–17) and Methods (pages 5–12) sections, respectively.

5. Majority of the paragraphs of the materials section lacks any references and that makes the research lacks reality. It must be taken in consideration.

*Authors’ Response: The Methods section (pages 5‒12) has been comprehensively revised to include detailed descriptions, including manufacturer names and locations, of all materials used in the study.

6. The results lack the significance data to be reliable.

*Authors’ Response: The Results and Figures now report statistically significant differences, where valid. In Figure 1, Figure 2, and Supplemental Figure 1 in S1 Appendix, asterisks indicate statistical significance at a level of P<0.05 versus LPS/vehicle, using the Mann-Whitney test.

7. The authors must interpret their results after its comparison with others in the discussion section.

*Authors’ Response: The Discussion section (pages 17–21) has been comprehensively revised to interpret the results in the context of similar studies and published literature.

8. The company names for any reagents must be mentioned in the materials and methods section.

*Authors’ Response: The Methods section (pages 5‒12) has been comprehensively revised to include detailed descriptions, including manufacturer names and locations, of all materials used in the study.

Reviewers’ Summary Comments

Reviewer #1: The authors evaluated the anti-inflammatory activity of lefamulin in a murine lipopolysaccharide-induced lung neutrophilia model. In spite of the scientific value of the subject, it lacks some important details. Therefore, it is greatly suggested that the manuscript is not ready to be accepted now. My decision is accept after major revisions. I have several comments listed below.

*Authors’ Response: We thank the reviewer for these comments and have revised the manuscript accordingly.

Reviewer #2: The manuscript has no significant data to prove their hypothesis. lefamulin is a very good antibiotic and can show anti-inflammatory responses. The authors failed to bring any conclusive decisions in it. The minimum dosage of drug used by them was high. The positive control used by them was 0.5 mg/kg. The compound used by them was 35 mg/kg BW. The author failed to show any cytokine quantification.

*Authors’ Response: We thank the reviewer for taking the time to review the manuscript. We have revised the manuscript to present data for a broad range of lefamulin and azithromycin doses and have clarified interpretation of the results in terms of how these doses correspond to clinically relevant exposure levels. In vivo and in vitro analyses of cytokines and chemokines have also been added. Accordingly, all sections of the current manuscript have been substantially revised.

---

## [Decision Letter · Decision Letter 1]

30 Jul 2021

Anti-inflammatory activity of lefamulin versus azithromycin and dexamethasone in vivo and in vitro in a lipopolysaccharide-induced lung neutrophilia mouse model

PONE-D-20-24953R1

Dear Dr. Gelone,

We’re pleased to inform you that your manuscript has been judged scientifically suitable for publication and will be formally accepted for publication once it meets all outstanding technical requirements.

Kind regards,

Partha Mukhopadhyay, Ph.D.

Section Editor

PLOS ONE

Additional Editor Comments (optional):

Reviewers' comments:

Reviewer's Responses to Questions

**Comments to the Author**

1. If the authors have adequately addressed your comments raised in a previous round of review and you feel that this manuscript is now acceptable for publication, you may indicate that here to bypass the “Comments to the Author” section, enter your conflict of interest statement in the “Confidential to Editor” section, and submit your "Accept" recommendation.

Reviewer #1: All comments have been addressed

Reviewer #3: (No Response)

2. Is the manuscript technically sound, and do the data support the conclusions?

Reviewer #1: Yes

Reviewer #3: Partly

3. Has the statistical analysis been performed appropriately and rigorously? 

Reviewer #1: Yes

Reviewer #3: I Don't Know

4. Have the authors made all data underlying the findings in their manuscript fully available?

Reviewer #1: Yes

Reviewer #3: Yes

5. Is the manuscript presented in an intelligible fashion and written in standard English?

Reviewer #1: Yes

Reviewer #3: Yes

6. Review Comments to the Author

Reviewer #1: (No Response)

Reviewer #3: (No Response)

7. PLOS authors have the option to publish the peer review history of their article (what does this mean?). If published, this will include your full peer review and any attached files.

Reviewer #1: No

Reviewer #3: No

---

## [Editor Report · Acceptance letter]

31 Aug 2021

PONE-D-20-24953R1 

Anti-inflammatory activity of lefamulin versus azithromycin and dexamethasone *in vivo* and *in vitro* in a lipopolysaccharide-induced lung neutrophilia mouse model 

Dear Dr. Gelone:

I'm pleased to inform you that your manuscript has been deemed suitable for publication in PLOS ONE. Congratulations! Your manuscript is now with our production department. 

Kind regards, 

on behalf of

Dr. Partha Mukhopadhyay 

Section Editor

PLOS ONE